# Adapting Neural Models with Sequential Monte Carlo Dropout

**Pamela Carreno-Medrano**    **Dana Kulić**    **Michael Burke**

Department of Electrical and Computer Systems Engineering
Monash University
Australia
`{pamela.carreno, dana.kulic, michael.g.burke}@monash.edu`

**Abstract:** The ability to adapt to changing environments and settings is essential for robots acting in dynamic and unstructured environments or working alongside humans with varied abilities or preferences. This work introduces an extremely simple and effective approach to adapting neural models in response to changing settings. We first train a standard network using dropout, which is analogous to learning an ensemble of predictive models or distribution over predictions. At run-time, we use a particle filter to maintain a distribution over dropout masks to adapt the neural model to changing settings in an online manner. Experimental results show improved performance in control problems requiring both online and look-ahead prediction, and showcase the interpretability of the inferred masks in a human behaviour modelling task for drone tele-operation.

**Keywords:** Model Adaptation, Online Robot Control and Prediction

## 1  Introduction

Neural models and policies are now ubiquitous in modern robotics. The prevailing approach to training these follows a two stage process – a large, comprehensive collection of data (often state and action pairs) is used to train a suitable model or policy, which is then frozen and deployed. Unfortunately, this results in models that are unable to adapt to changes in the environment, which is a particular concern in robotics. For example, it would be preferable for a robot dynamics model to handle context dependent kinematic or dynamic properties, or a collaborative robot relying on predictions of human behaviour to adapt to different human abilities or preferences. Many existing adaptive control techniques [1] attempting to tackle this problem rely on carefully considered parametric models, but these may lack the requisite capacity for prediction that is typically associated with neural models. In contrast, meta-learning and adaptive neural control approaches addressing this problem are often quite cumbersome to train and implement.

This paper introduces a simple and effective approach to achieve adaptation for neural network models. First we train a model, $f_{\boldsymbol{\theta}}(\cdot)$ using dropout [2], on data gathered from different settings or tasks. Dropout can be considered analogous to randomly masking neurons in the network, producing an ensemble of predictive models, or a distribution over possible outputs [3]. At run-time, a particle filter is used to update the dropout mask, depending on the predictive quality of the model, so as to condition the network to operate in a given context. This allows for neural models that can easily and rapidly adapt to changes in the environment or behaviour of a human collaborator in online settings.

Results show that this simple-to-implement approach improves upon a more complex first-order meta-learning training strategy, and can alleviate potential failures due to catastrophic forgetting. Moreover, we show that the resulting masks can be interrogated to obtain additional information about the adaptations required for particular contexts. We also show how this information can be leveraged in a human modelling application to answer questions about a given user. In summary, the core contributions of this work are to show that: **(1)** sequential Monte Carlo dropout is an effective technique for online model adaptation and control in rapidly changing settings, and **(2)** the masks

6th Conference on Robot Learning (CoRL 2022), Auckland, New Zealand.

inferred by this process provide a context-dependent 'signature' that is of particular value for human behaviour modelling in support of personalised robot decision making.

## 2 Background and Related Work

**Model adaptation**: Model adaptation in an online fashion is a core focus of adaptive control [1], and typically involves the design of a suitable observer or estimation scheme [4] to infer parameters of interest. Many of these system identification approaches [5, 6, 7, 8, 9] rely on models conditioned on known physical properties like friction, mass and gravity, and infer these from data. These approaches require knowledge of the parametric form of uncertainty and are only feasible for very specific model structures [10]. Our approach introduces an adaptation mechanism applicable to more general predictive neural models.

For deep learning, model adaptation can be studied as a generalisation problem in a meta-learning context. Popular approaches like model agnostic meta-learning (MaML) [11] seek to learn network weights that can be easily adapted for different tasks by gradient descent via specialised training regimes that alternate between task specific training and maintaining a representation that is well suited to multiple tasks. Although time-consuming, faster, first-order meta-learning approximations such as Reptile [12] have proven relatively effective. The idea of enhancing generalisation by learning well-structured internal representations underpins a number of works such as ProtoNet [13] and Relation networks [14], alongside more recent approaches that rely on multi-task training with an explicit learned auxiliary input [15, 16] that can be adapted online. Our approach can also be viewed through a representation learning lens, in that masks 'prime' the network to produce context dependent outputs. This representation-priming perspective also aligns with recently developed transformer-based architectures tackling meta-learning (e.g., [17, 18, 19]). Here, key embeddings are used to unlock network components required for context specific prediction. The dropout mask used in our work is analogous to a transformer key that is inferred at run time.

Several works targeting online model adaptation have argued that we should ideally update our models and beliefs in response to evidence in a Bayesian fashion. This approach is taken in elastic weight consolidation, which aims to reduce the plasticity of network weights that are important to a task [20]. There is also some evidence that MaML can be reinterpreted as a Bayesian hierarchical model [21, 22]. Dropout [2] is a popular regularisation technique for neural network training, and has also been used to represent prediction uncertainty in deep learning [3]. We leverage dropout within a sequential importance sampling framework to provide an explicit Bayesian update rule for mask inference and model adaptation. Like [20], this preserves connections important to a task.

As an alternative to back-propagation, evolutionary, sampling-based approaches to training neural models [23, 24, 25] have been explored and shown to be effective in a reinforcement learning context [26]. Similarly, the cross entropy method is a popular technique for global sampling-based optimisation and planning [27, 28]. Pathnet [29] uses evolution to find the neural network pathways that should be trained for specific tasks. Particle filtering has also been used to prune neural networks for compression [30]. Our work is similar, but considers the online adaptation case and samples dropout masks to infer a distribution over masks, rather than considering all weights and connections.

**Human modelling**: The dropout mask inferred at run-time using our approach can be considered as an embedding or summary of the context-specific adaptation required by the network at a given time step. This is valuable for adaptive human modelling in shared-autonomy and similar collaborative applications, where context-specific masks could help guide the type of assistance offered to an operator. Human behaviour models help robots to understand human preferences, predict future human actions, and adapt to changes in human behaviour. Two general approaches to human behaviour modelling are the most common in robotic applications. The first postulates that humans are rational agents whose actions are in accordance with some (un)known intentions and rewards [31]. To account for human variability and diversity in behaviour – in particular deviations from optimal action choices, these models are augmented with computational interpretations of high-level decision factors such as trust [32], expertise [33], fatigue [34], and adaptability [35], among others. These factors can be inferred in an online manner during interaction, thus allowing robots to adapt their responses to different human users (e.g., [36]). Often preferred because of their simplicity and interpretability, these models suffer two main drawbacks. First, they strongly rely on simplistic parametric model structures, which can limit model expressivity and predictive power. In particular,

most models assume that a single high-level decision factor is sufficient to capture and explain most of the observed human behaviour (e.g., [36]). Second, to make inference tractable in real-time, these approaches are often limited to simplified state and action space representations, and further assume that the model parameters remain stationary for the duration of the whole interaction (e.g., [35]). Our approach aims to relax these constraints by introducing an adaptation mechanism that can be used with more general predictive and expressive neural models and that also captures changing behaviour.

In the second family of approaches, collected human behaviour data is used to train machine learning models that predict future behaviour (i.e., imitation learning) [37, 38, 39]. Because of their data-driven nature and assuming good quality data is provided during training, these models can implicitly capture most of the sources of variability in the observed human behaviour without making explicit assumptions about what those sources are. Thus, these models of human behaviour are no longer limited to single decision factors. However, this increase in model expressivity and predictive power comes at the cost of limited interpretability [40]. Furthermore, these models can fail to generalize and adapt to humans whose behaviour significantly differ from the training data set [16]. Our approach addresses the interpretability and adaptation limitations of imitation-based human models through the online estimation of dropout masks. As shown in Sec. 5, this mask inference process allows for successful local adaptation in extrapolation (to human users never seen during training). Similarly, we show that contextual information pertaining to high-level factors such as a human operator's skill level and preferences can be queried through a simple k-nearest mask approach.

## 3   Method

We consider a standard Markov decision process, with $\mathbf{x}_t$ denoting a state at time $t$, and $R_t$ denoting the reward received at time $t$ after transitioning from state $\mathbf{x}_{t-1}$ to state $\mathbf{x}_t$, due to control action $\mathbf{u}_t$. Let $f_\theta(\cdot)$ be a neural network with parameters $\theta$. Networks such as these are frequently trained to approximate dynamics models for future state prediction $\mathbf{x}_t \sim f_\theta(\mathbf{x}_{t-1}, \mathbf{u}_t)$, for reward prediction $R_t \sim g_\theta(\mathbf{x}_{t-1}, \mathbf{u}_t)$, or as policies for selecting an action given a state $\mathbf{u}_t \sim \pi_\theta(\mathbf{x}_t)$.

Our goal is to find a way of adapting a pre-trained network to changing settings in response to some deviation between predicted states, actions or rewards and those actually observed. Let $f_\theta(\cdot|\mathbf{M}_t)$ denote a neural network trained using dropout regularisation. Here $\mathbf{M}_t$ represents a binary mask which suppresses or drops neurons in the network. When training with dropout, neurons are randomly masked during the forward pass. We do not consider any specialised training regime, and simply use multi-task training - samples drawn at random from a large training dataset.

We now show how this can be exploited to enable online adaptation. For convenience, we restrict the discussion below to a dynamics model setting where a neural network is trained to predict a state $\mathbf{x}_t$, given a previous state $\mathbf{x}_{t-1}$ and action $\mathbf{u}_t$. However, it should be noted that the proposed approach can be used to adapt policies, reward predictors and in many other settings.

### 3.1   Online Mask Adaptation

We use the deviation between predicted $\mathbf{x}_t$ and actual observations $\mathbf{z}_t$ to form a Gaussian likelihood,

$$p(\mathbf{z}_t|\mathbf{M}_t) = \mathcal{N}(\mathbf{z}_t|f_\theta(\mathbf{x}_{t-1}, \mathbf{u}_t|\mathbf{M}_t), \boldsymbol{\Sigma}), \tag{1}$$

where $\boldsymbol{\Sigma}$ denotes measurement noise. This likelihood measures the chances of making an observation given a dropout mask $\mathbf{M}_t$. Taking an ensemble view of model training with dropout, we aim to find the mask, and corresponding model, that results in the most accurate prediction in a given setting. Formally, our goal is to determine a posterior distribution over dropout masks $p(\mathbf{M}_t|\mathbf{z}_{1:t})$ conditioned on a series of observations $\mathbf{z}_{1:t}$. To do so, we formulate a sequential update rule

$$p(\mathbf{M}_t|\mathbf{z}_{1:t}) \propto p(\mathbf{z}_t|\mathbf{M}_t)p(\mathbf{M}_t|\mathbf{z}_{1:t-1}), \text{ where} \tag{2}$$

$$p(\mathbf{M}_t|\mathbf{z}_{1:t-1}) = \int p(\mathbf{M}_t|\mathbf{M}_{t-1})p(\mathbf{M}_{t-1}|\mathbf{z}_{1:t-1})\mathrm{d}\mathbf{M}_{t-1}, \tag{3}$$

using an appropriate transition model $p(\mathbf{M}_t|\mathbf{M}_{t-1})$ for the dropout mask. In this work, we propose to randomly bit-flip some small fraction ($d\,\%$) of the mask for the transition model, $p(\mathbf{M}_t|\mathbf{M}_{t-1})$. This allows for some exploration over potential new masks and adaptation to new settings while

keeping existing masks close to those associated with networks best suited to current settings. Importantly, bit flipping can be efficiently computed using an xor operation.

Since the integral in (3) is intractable, we approximate it using importance sampling and hence reduce the computation of the posterior over masks to a bootstrap particle filter [41]. That is, we approximate the distribution over masks using a set of $N$ weighted particles, $p(\mathbf{M}_t|\mathbf{z}_{1:t}) \approx \sum_{i=1}^{N} w_i \delta(\mathbf{M}_t^i)$, with weights $w_i \propto p(\mathbf{z}_t|\mathbf{M}_t^i)$, particle $\mathbf{M}_t^i$ drawn from the transition distribution $p(\mathbf{M}_t^i|\mathbf{M}_{t-1}^j)$, and Dirac measure $\delta(\cdot)$ at point $(\cdot)$. A resampling step is used after each iteration to prevent particle degeneracy. This produces a distribution over masks and a corresponding posterior over the predicted state

$$p(\mathbf{M}_t|\mathbf{z}_{1:t}) \approx \frac{1}{N} \sum_{i=1}^{N} \delta(\mathbf{M}_t^i); \tag{4}$$

$$p(\mathbf{x}_t|\mathbf{x}_{t-1}, \mathbf{u}_t, \mathbf{z}_{1:t}) = \int p(\mathbf{x}_t|\mathbf{x}_{t-1}, \mathbf{u}_t, \mathbf{z}_{1:t}, \mathbf{M}_t)p(\mathbf{M}_t|\mathbf{z}_{1:t})\mathrm{d}\mathbf{M}_t$$

$$\approx \frac{1}{N} \sum_{i=1}^{N} \delta\left(f_\theta(\mathbf{x}_{t-1}, \mathbf{u}_t, \mathbf{M}_t^i)\right). \tag{5}$$

For downstream tasks like control, where we are only interested in the adapted prediction model, we compute a minimum-mean squared error mask estimator

$$\bar{\mathbf{M}}_t = \frac{1}{N} \sum_{i=1}^{N} \mathbf{M}_t^i, \tag{6}$$

from the mask particles and use this to condition the network $f_\theta(\cdot)$. We note that since the likelihoods are bounded and we use a standard resampling scheme, convergence of the proposed approach is independent of the state dimension, with the rate of convergence in $\frac{1}{N}$ [42]. The filter also allows for variable particle sizes, which can be increased or decreased online depending on the effective particle size as defined by $N_{eff} = \frac{1}{\sum_{i=1}^{N}(w^i)^2}$ [43]. Algorithm 1 illustrates the adaptation process, which we term sequential Monte Carlo dropout (SMCD). We first sample $N$ possible masks from the prior mask distribution by bit flipping, and then make a batch prediction of possible states for each mask. We then compute the likelihood for each mask using (1), normalise

---

**Algorithm 1** Sequential Monte Carlo Dropout (SMCD) Model Adaptation

**Input:** Trained net $f_\theta$, Observation sequence $\mathbf{z}_{1:T}$, Transition probability $d$, Measurement uncertainty $\boldsymbol{\Sigma}$
**Output:** Mask estimate sequence $\bar{\mathbf{M}}_{1:T}$
Initialise $N$ random masks $\{\mathbf{M}_0^i\}_{i=1...N}$
**for** $t = 1 \dots T$ **do**
    **for** $i = 0 \dots N$ **do**
        Sample $\mathbf{M}_t^i \sim p(\mathbf{M}_t^i|\mathbf{M}_{t-1}^i)$ by bit-flipping
        Sample $\mathbf{x}_t^i = f_\theta(\mathbf{x}_{t-1}, \mathbf{u}_t|\mathbf{M}_t^i)$
        Evaluate $w^i = \mathcal{N}(\mathbf{z}_t|\mathbf{x}_t^i, \boldsymbol{\Sigma})$
    **end for**
    Draw $N$ masks: $\{\mathbf{M}_t^i\}_{i=1...N} \sim \frac{\sum_{i=1}^{N} w_i \delta(\mathbf{M}_t^i)}{\sum_{i=1}^{N} w_i}$
    Compute best mask estimate $\bar{\mathbf{M}}_t = \frac{1}{N} \sum_{i=1}^{N} \mathbf{M}_t^i$
**end for**

---

and re-sample, to obtain an approximate distribution over masks that gives predictions close to the observation. Finally, we compute a minimum-mean squared error mask, which can be used to condition the network in downstream tasks. The process repeats recursively and online.

A key benefit of this approach is that it does not change the underlying models learned. The original prediction can be recovered at any time, by simply restoring or resampling the mask. This is in contrast to approaches that continue to train or take gradient steps in response to errors, which may be suitable for continual learning, but are vulnerable to catastrophic forgetting [44, 20].

## 4 Experiments

We investigate model adaptation using SMCD both in simulation and with real-world human data. In simulation, we use a two-link arm with changing link lengths and a 7 Degree of Freedom (DOF)

Panda arm reaching task with varying end-effector length. We evaluate the performance of the proposed method in prediction and control using the the two-link arm task and 7-DOF Panda reaching task respectively. The real-world data corresponds to a drone tele-operation scenario where operators of varying skill levels completed several landing tasks. This scenario illustrates mask interpretability by showing how different high-level characteristics (an operator's landing strategy and skill level) can be determined by a simple mask comparison process. We refer the reader to the Appendix for additional experiments including applications to Reinforcement Learning (RL) settings.

## 4.1 Two-Link Arm Task (Forward Kinematics)

The 2-link arm follows the dynamics and forward kinematics:

$$\dot{q}_1 = u_1, \ \dot{q}_2 = u_2 \tag{7}$$
$$x = l_1 \cos(q_1) + l_2 \cos(q_1 + q_2), \ y = l_1 \sin(q_1) + l_2 \sin(q_1 + q_2). \tag{8}$$

We train neural models $f_\theta(\boldsymbol{q})$ to perform forward kinematics (predicting end-effector position $\mathbf{x} = [x, y]^\mathrm{T}$ given joint angles $\boldsymbol{q} = [q_1, q_2]^\mathrm{T}$) for variable limb lengths $l_1, l_2$. Training data is gathered by randomly initialising $q_i \sim U(-\pi, \pi)$, and episodically motor babbling $u_i \sim \mathcal{N}(0, 1)$ for 10 steps. We generate 150 episodes for 1000 tasks, with fixed link lengths $l_i \sim \mathcal{N}(1, 0.3)$ in each task.

The two-link forward kinematics task is used to investigate the adaptation properties of SMCD in $N$-step look-ahead prediction after $T$ adaptation steps[1]. Specifically, we evaluate prediction error for an unknown manipulator with link lengths ($l_i \sim \mathcal{N}(1, 0.3)$) fixed over a motor babbling episode ($u_i \sim \mathcal{N}(0, 1)$), starting from a randomly sampled angle $q_i \sim U(-\pi, \pi)$. This evaluates the "interpolation" ability and suitability for receding-horizon model predictive control tasks. We benchmark against an oracle model, no adaptation, a latent variable model, and a model trained using Reptile [12], a first order meta-learning approach, and investigate adaptation using both gradient descent and SMCD.

## 4.2 7-DOF Panda Arm Reaching Task (Manipulator Control)

The 7-DOF Panda manipulator is simulated using the python robotics toolbox [45] with a tool that introduces an arbitrary unknown end-effector extension ($l \sim U(-0.5, 0.5)$). As with the two-link arm task, we train neural models $f_\theta(\boldsymbol{q})$ to perform forward kinematics (predicting end-effector position $\mathbf{x} = [x, y, z]^\mathrm{T}$ given joint angles $\boldsymbol{q} = [q_1, \ldots, q_7]^\mathrm{T}$). Our goal is to drive the Panda from a fixed reference pose to within 5 cm of a randomly selected and kinematically feasible end-effector goal $\mathbf{x}_g$ in less than 200 time steps. To reach the goal, we use the proportional-derivative (PD) control law $\mathbf{u} = -K_1 \mathbf{J}^\dagger (\mathbf{x} - \mathbf{x}_g) - K_2 \dot{\boldsymbol{q}}$, where $\mathbf{J}^\dagger$ corresponds to the Jacobian of the end-effector position ($\mathbb{R}^3$) joint with respect to the angles ($\mathbb{R}^7$). We consider the case where measurement of the true end effector position is only available every $n-$th timestep, which requires model-reference control when measurements are not available. This measures the ability of the adapted model to generalise to global context changes under a realistic control setting in which observations are captured at a lower rate than the required controller frequency.

## 4.3 Drone Landing Task (Mask Interpretability)

We use the human data previously introduced in [46] for this task[2]. This dataset comprises 528 tele-operation trajectories collected for 33 human operators (16 trajectories per operator) with different skill levels in a simulated environment. Here, operators were asked to pilot and land a drone on one of five platforms by providing linear velocity commands through keyboard inputs. As shown in Fig 1a, the operator's view was fixed to a single-point perspective, which resulted in an increase in task complexity the further away the target platform was located. The simulated environment and landing task sequence were the same for all operators.

We train neural models $\boldsymbol{u}_t \sim f_\theta(\boldsymbol{x}_t)$ to predict the operator's action $\boldsymbol{u}_t$ given the current state $\boldsymbol{x}_t$ for variable tasks (i.e., different target platforms), user skill levels, and landing strategies (see Fig. 1b for examples of tele-operation trajectories with these characteristics). A tele-operation trajectory

---

[1]Results showcasing SMCD's performance in online, on-policy adaptation using a proportional-derivative (PD) control law for this task are shown in Appendix.

[2]Secondary analysis of this data was reviewed and approved by Monash University Research Ethics Committee.

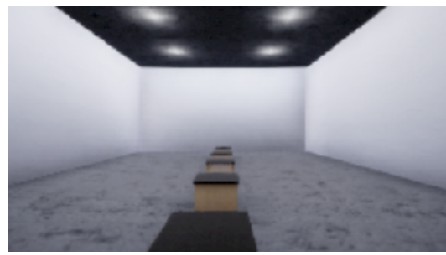

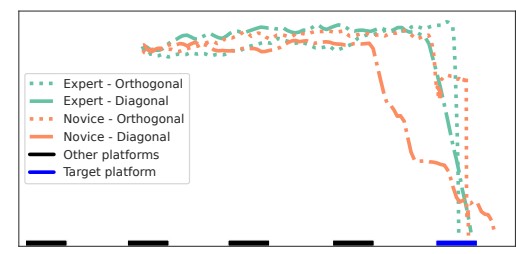

(a) Drone landing simulation environment.          (b) Examples of tele-operation trajectories (YZ view).

Figure 1: Drone landing data consists of tele-operation trajectories collected in a simulated environment (a). Data includes trajectories executed by operators of varying skill (one novice and one expert) and captures different landing strategies (orthogonal or diagonal) as shown in (b).

$\boldsymbol{\tau} = (\boldsymbol{x}_1, \boldsymbol{u}_1, \ldots, \boldsymbol{x}_T, \boldsymbol{u}_T)$ consists of a sequence of states and actions, where each state $\boldsymbol{x}_t = (\mathbf{p}_t, \mathbf{v}_t, d_t)$ denotes the drone's 3D position $\mathbf{p}_t = (x_t, y_t, z_t)$, linear velocity $\mathbf{v}_t = (\dot{x}_t, \dot{y}_t, \dot{z}_t)$, and distance $d_t = |\mathbf{g}^i - \mathbf{p}_t|_2$ to target platform $\mathbf{g}^i$ at time $t$; and $\boldsymbol{u}_t = (\dot{x}_t, \dot{y}_t, \dot{z}_t)$ corresponds to the linear velocity commands provided by the human operator. We randomly split the dataset into training (83%), known-operators testing (10%), and unknown-operators testing (7%) sets.

To investigate the interpretability of the masks obtained using SMCD we evaluate whether one high-level characteristic of a trajectory (landing strategy) and one characteristic about the operator (skill level) are captured by the inferred masks[3] (see Fig. 1b for trajectories showcasing these characteristics). Specifically, we measure how the prediction confidence of each target characteristic changes as the number of observed state-action pairs seen during mask inference increases. We use the distance-based confidence score proposed in [47] to evaluate this. Given the mask estimate, $\bar{\mathbf{M}}_{1:n}$, for trajectory $\boldsymbol{\tau}$ obtained after $n$ observations, with $1 \leq n < T$, let $A(\bar{\mathbf{M}}_{1:n}) = \{\bar{\mathbf{M}}_{\text{train}}^j\}_{j=1}^k$ denote the set of $k$-nearest masks of $\bar{\mathbf{M}}_{1:n}$ in the training set, and let $\{c^j\}_{j=1}^k$ denote the corresponding characteristic labels associated with masks in $A(\bar{\mathbf{M}}_{1:n})$. Following [47], the confidence score $D(\boldsymbol{\tau})$ for the assignment of trajectory $\boldsymbol{\tau}$ to characteristic $\hat{c}$ based on the mask estimate $\bar{\mathbf{M}}_{1:n}$ is defined as:

$$D(\boldsymbol{\tau}) = \frac{\sum_{j=1, c^j = \hat{c}}^k e^{-\|\bar{\mathbf{M}}_{1:n} - \bar{\mathbf{M}}_{\text{train}}^j\|_2}}{\sum_{j=1}^k e^{-\|\bar{\mathbf{M}}_{1:n} - \bar{\mathbf{M}}_{\text{train}}^j\|_2}}. \tag{9}$$

The score $D(\boldsymbol{\tau}) \in (0, 1)$ (the higher the better) is monotonically related to the local density of trajectories in the training set that share similar characteristics and lie in the neighborhood of trajectory $\boldsymbol{\tau}$, as defined by the distance between inferred masks.

## 5 Experimental Results

**Two-link arm: look-ahead prediction results**: Fig. 2 shows the root mean square prediction error (RMSE) over 1000 trials for varied burn in steps and prediction horizons (identical for all methods compared). We compare best performing models[4] trained using multi-task pre-training (M) and Reptile meta-learning (R), and then adapted using either SMCD (S), gradient descent (G), or no adaptation (N). We also include an oracle model baseline (O) where link-length parameters in a known forward kinematics model are inferred directly using a particle filter and a latent variable model (LV) where task variability and contextual information are both captured using the low-dimensional hidden embedding, $h \in \mathcal{R}^8$, of a variational recurrent auto-encoder (VRNN) [48]. At test time, burn-in steps are used to update the VRNN's hidden embedding later used to predict each step of the look-ahead horizon[5].

Multi-task training with SMCD is most effective (M+S), with the exception of the oracle model. Interestingly, adaptation using gradient descent seems to exhibit more local adaptation, as prediction

---

[3]Characteristic definitions and details are provided in the Appendix

[4]Best performing models were selected following a hyperparameter search - see Appendix

[5]Since neither the multi-task nor the Reptile model networks are recurrent models, the VRNN model was constrained to train using sequences of length 2 to make the comparison between models possible.

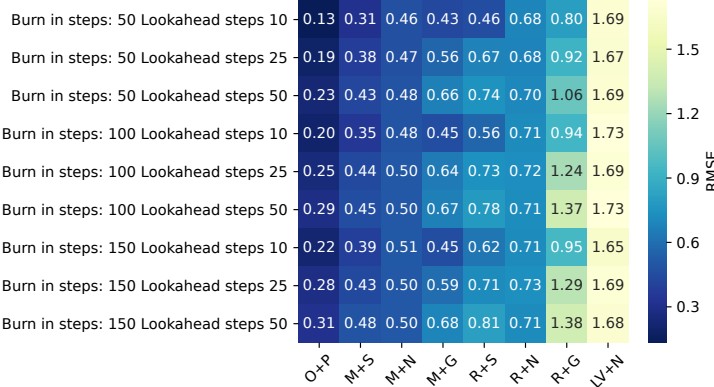

Figure 2: Look-ahead prediction errors. (O: Oracle, M: multi-task, LV: Latent Variable, R: Reptile; P: Particle Filter, S: SMCD, G: Gradient descent, N: No adaptation.)

error grows larger with look-ahead distance than when SMCD is applied. Local adaptation seems to provide an accurate model in a specific region of state space, rather than globally adapting to a particular link length. The Reptile model performs surprisingly poorly, particularly given that it was trained for substantially longer (1000 epochs) than the multi-task model (200 epochs). A similar performance is observed for the latent variable model.

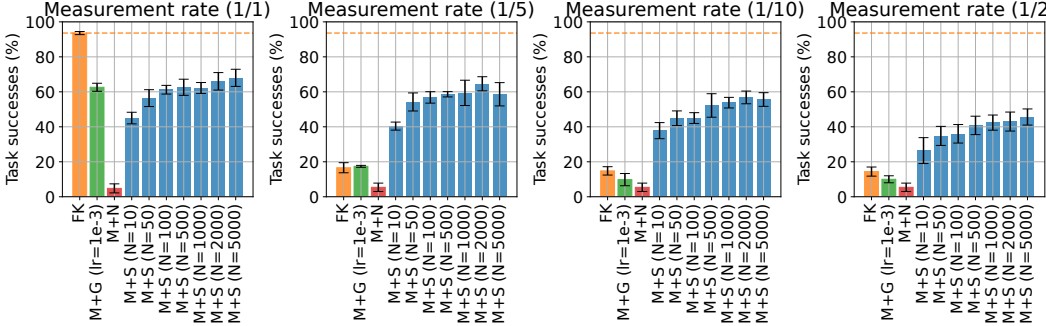

Figure 3: 7-DOF Panda manipulator control over increasing horizons. (M: multi-task, FK: Analytical, G: Gradient descent, S: SMCD, N: No adaptation) The orange line indicates best case performance when true measurements are available at every time step.

**7-DOF Panda manipulator: online control results**: Fig. 3 shows the success rates in the adaptive Panda control task for decreasing measurement rates. We used a 3-layer, 1024 dimensional fully connected network with ReLU activations to learn forward kinematics models. The model was trained with 100000 sampled joint angles and end-effector position combinations, with an arbitrary tool offset. A total of 100 reaching tasks were completed during testing and results averaged across 5 seeds.

We compare an analytical model (FK) that fails to account for the tool offset and a multi-task model with no adaptation (M+N), adapted using gradient descent (M+G), and adapted using SMCD (M+S). When true measurements are available at every time step, FK represents an oracle model that only fails due to singularity effects in the control law. M+G performance degrades significantly as the rate of measurement decreases and the model is required to be used for longer control periods. When selecting the M+G learning rate, we found that this performance degradation increases with learning rate, which appears to indicate that gradient descent is only suitable to adapt representations to very local contexts, and the model is no longer useful in other regions of the state space. In contrast, M+S appears to be much more effective at capturing global context, although performance still degrades when the model is used over longer periods with no feedback. Increasing the number of particles results in a moderately improved performance.

**Drone landing task: mask interpretability results**: Fig. 4 shows the average confidence score obtained for the prediction of the true skill level of a trajectory's operator (i.e., novice, intermediate, or expert) and true landing strategy (i.e., orthogonal or diagonal) for all trajectories in our testing sets as the number of time steps used during mask inference increases.

Overall, we observe relatively good average confidence scores ($\sim$ 0.7) for both testing sets. This suggests that the masks inferred using SMCD contain enough global information to allow for the identification of operator skill level and the landing strategy. Both characteristics can be identified early on in the trajectory. This is particularly important for the landing strategy, which appears to change more towards the end of a trajectory. Better confidence scores are obtained for trajectories executed by unknown operators than for the known operator testing instances. This might be due to imbalances in the number of instances that belong to each possible characteristic,

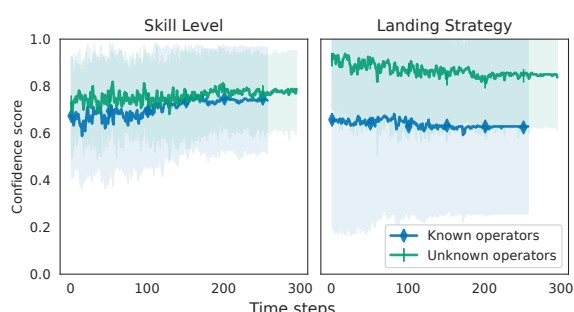

Figure 4: Average confidence score for the correct assignment of the true skill level and landing strategy for all testing trajectories (1-sigma shaded traces). Results are presented for the set of 5-nearest masks.

as visual inspection of trajectories indicated unknown operators exhibited trajectory characteristics more common in the dataset. Additional results on mask interpretability and look-ahead prediction for this task are provide in the Appendix.

## 6 Limitations

Although neural dynamics modelling holds the potential to scale to high dimensional inputs of varied modalities, this work has mostly considered adaptation in proprioceptive state spaces. Future work should explore the potential of the proposed approach to operate in models using higher dimensional image inputs. It should be noted that control with adaptive neural models may lack the stability and safety guarantees associated with conventional dynamics modelling. Moreover, although SMCD is fast, and converges rapidly, doing so does require an energy expensive GPU. While all of the models considered in this work can be fit on a small size embedded GPU, users of the neural adaptation schemes proposed here should carefully consider their model size to determine whether sampling or back-propagation is more computationally efficient, given the relative efficacy in the multi-task training setting. Given the nature of representations learned by neural networks, this is likely to be task dependent. However, as shown above, model adaptation using SMCD can identify and adjust to contexts and unlock neural representations highly suited to model-based control in settings where gradient based adaptation schemes can result in catastrophic forgetting.

## 7 Conclusion

This work proposed a simple neural network adaption strategy, relying on sequential Monte Carlo adaption of the masks in a dropout regularisation network layer. These masks offer a level of interpretability, capturing both local and some global contextual information, as illustrated in a real world human behavioural modelling application. Results show that when paired with multi-task training, this simple adaptation strategy outperforms more computationally complex meta-learning schemes and overcomes some of the catastrophic forgetting problems associated with gradient descent in online adaptation settings.

**Acknowledgments**

We thank the reviewers for their useful comments and valuable recommendations.

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
