# OpenReview forum: "Adapting Neural Models with Sequential Monte Carlo Dropout"
_robot-learning.org/CoRL/2022/Conference — CoRL 2022 Poster_

### Official Review · Reviewer_YMaq · 2022-07-25

**Originality:** Good
**Technical Quality:** Good
**Clarity Of Presentation:** Good
**Impact:** 4

**Recommendation:**

Weak Accept: I recommend accepting the paper, but will not argue for my recommendation if the majority of other reviewers have a different opinion.

**Summary:**

This paper presents a method to perform system identification with with a world model by leveraging dropout masks.  The general idea is to train a neural network on a randomized version of the system dynamics, sampled from a domain encompassing that which is likely to be seen in the final system.  This network is trained with dropout, using no particular training regime, therefore re-sampling a dropout mask at each forward pass.  Then, during use, there is a 'Mask Adaptation' phase. This is similar in spirit to CEM, and samples a set of random masks, evaluates which ones provide the best next-state estimation, and then re-samples masks around the space of well-performing masks from the previous iteration to fine-tune the mask selection.  This process continues permanently, allowing for not only initial sys-id, but also on-line adaptation to non-stationarity in the system dynamics.

**Issues:**

The main issues are described above: the mathematical description of the algorithm is a bit awkward, and could perhaps be better contextualized relative to existing approaches (EM, CEM).  The other main issue is the relatively simple and low-impact experimental section which really misses the opportunity to make a compelling case for robotics control in higher-dimensional systems.  I nevertheless argue for a weak accept because I find the suggested approach very appealing, but will not defend it if other reviewers find the experimental section lacking.

**Quality Of The Limitations Section:**

Additional details required

**Reviewer Expertise:**

3: The reviewer is fairly confident that the evaluation is correct

**Robotics Focus:**

Highly relevant to robotics but no hardware experiments

**Strengths And Weaknesses:**

I find the idea very clever, simple, and easy to implement.  It makes intuitive sense that this could work well, and it provides an appealing approach to dealing not only with sys-id but also adaptivity to non-stationarity in the system dynamics.  These are both very real prbolems in physical systems and having a simple, stable, easy-to-implement solution is very appealing.

The main downside of the paper for me is the very limited experimental results.  I would have really liked to see this approach in use in more controls-oriented tasks, on higher-dimensional systems.  There exist very easy to use environments such as DM Control, Gym or Brax that can be easily perturbed (or even more specific manipulation environments such as Robosuite), and existing model-based controllers such as PDDM, or various MPPI implementations which could easily be extended with this adaptive world-model.  Experiments in these domains would be much more convincing in demonstrating this approach's practical abilities in higher-dimensional control tasks.

I'm also not convinced the formal description of the approach in Eq. 1-3 was particularly heplful, I found it rather more confusing and only understood the mechanism by reading Alg. 1.  I would encourage the authors to perhaps provide a closer comparison to CEM, perhaps the algorithm could be cast as a CEM variant which would also make it easier to contextualize and describe.

**Summary Of Recommendation:**

I think the core idea of the paper is interesting and applicable to a large number of domains.  It feels so intuitive that I wonder if it has been published elsewhere in a different form, but I wasn't able to find a paper suggesting an explicit adaptation phase for dropout-induced variability in the network outputs.  The paper correctly (to the extent of my knowledge) references existing work in dropout approaches to bayesian inference.

---

> ### Author Response · Authors · 2022-08-18
> **Response to Reviewer YMaq**
>
> Thank you to the reviewer for their valuable comments and suggestions and for recognising the importance of this work and the strength of the proposed approach. We briefly respond to reviewer queries below and are currently running additional ablations and experiments to address concerns around experimental support.
>
> **The main downside of the paper for me is the very limited experimental results. I would have really liked to see this approach in use in more controls-oriented tasks, on higher-dimensional systems. There exist very easy to use environments such as DM Control, Gym or Brax that can be easily perturbed (or even more specific manipulation environments such as Robosuite), and existing model-based controllers such as PDDM, or various MPPI implementations which could easily be extended with this adaptive world-model. Experiments in these domains would be much more convincing in demonstrating this approach's practical abilities in higher-dimensional control tasks.**
>
> We are currently running additional experiments on a broader suite of control tasks to better showcase the efficacy of the proposed approach.
>
> **I'm also not convinced the formal description of the approach in Eq. 1-3 was particularly heplful, I found it rather more confusing and only understood the mechanism by reading Alg. 1. I would encourage the authors to perhaps provide a closer comparison to CEM, perhaps the algorithm could be cast as a CEM variant which would also make it easier to contextualize and describe.**
>
> Thanks for pointing this out, we have referenced CEM (see line 72 in the main text). However, the particle filter formulation is different to CEM, and allows us to draw on some useful results from the statistics community, which we have now included to stress the value of this approach. To summarise, CEM is a global optimisation approach tailored for rare event sampling, and a core difference is that the proposal distribution is typically a Gaussian with mean and variance adapted based on some high-value prior samples. In contrast, a particle filter is a sequential update rule based on a fixed proposal distribution (in our case the bitflip mask dynamics model) and provides important convergence guarantees to the tracked state (in our case the masks). Of particular value here are convergence results that break the curse of dimensionality [2] (convergence is independent of state dimensionality, and the rate of convergence is 1/N, where N here is the batch size or number of masks), along with known approaches to dynamically choose the number of particles [3] based on the effective sample size (Neff ~ 1/sum (wi^2)). We have added this context  to the paper to better motivate this choice of formulation.
>
> [2] D. Crisan and A. Doucet, "A survey of convergence results on particle filtering methods for practitioners," in IEEE Transactions on Signal Processing, vol. 50, no. 3, pp. 736-746, March 2002, doi: 10.1109/78.984773.
> [3] Kong, A, Liu, J.S. and Wong, W. H. (1994) Sequential imputations and Bayesian missing data problems. Journal of the American Statistical Association, 89, 1032–1044.
>
> **The other main issue is the relatively simple and low-impact experimental section which really misses the opportunity to make a compelling case for robotics control in higher-dimensional systems.**
>
> We are currently running additional experiments on a broader suite of control tasks to better showcase the efficacy of the proposed approach in higher-dimensional systems.

---

> > ### Comment · Reviewer_YMaq · 2022-08-23
> > **Response to discussion**
> >
> > Thank you for your response, the additional context on the choice of particle filtering vs. CEM was helpful.  I am looking forward to the more extensive experimental results.

---

> > > ### Author Response · Authors · 2022-08-27
> > > **Additional robotic experiments (Panda Arm)**
> > >
> > > We have added experiments on a 7-DOF Panda arm. This experiment provides important insights into the practical applicability of SMCD in real-time control. In this experiment, a 7-DOF Panda manipulator is simulated with a tool that introduces an arbitrary unknown end-effector extension. Our goal is to drive the Panda from a fixed reference pose to within 5 cm of a randomly selected and kinematically feasible goal within 200 time-steps, using a PD control law and the Jacobian of the end-effector position joint with respect to the angles. We consider the case where measurement of the true 7-DOF Manipulator Control (Panda) end effector position is only available every nth timestep, which requires model-reference control when measurements are not available. This measures the ability of the adapted model to generalise to global context changes under more realistic control settings, where observations are captured at a lower rate than the required controller frequency.
> > >
> > > The success rates in the adaptive Panda control task for decreasing rates of measurement are shown in the attached image. FK is an analytical model that fails to account for the tool offset. When true measurements are available at every time step, this represents an oracle model and the only failures here are due to singularity effects in the control law. M+G is a multi-task model adapted using gradient descent, M+N is the same model with no adaptation, and M+S is the same model adapted using SMCD. M+G performance degrades significantly as the rate of measurement decreases and the model is required to be used for longer control periods. When selecting the M+G learning rate, we found that this performance degradation increases with the learning rate, which appears to indicate that gradient descent is only suitable to adapt representations to very local contexts, and the model is no longer useful in other regions of the state space. In contrast, SMCD appears to be much more effective at capturing global context, although performance still degrades when the model is used over longer periods with no feedback. Increasing the number of particles results in moderately increased performance.

---

> > > > ### Author Response · Authors · 2022-08-27
> > > > **Additional experiments (continuous control in high-dimensional spaces)**
> > > >
> > > > We also consider a continuous control task from the OpenAI Gym suite which provides important insights into SCMD's scalability to higher dimensions and extrapolation to unseen environments. In this task, the goal is to make a four-legged robot move forward as quickly as possible. In this task, the robot's observed state $o_t$ (which includes the torso position and orientation, joint angles, the torso linear and angular velocity, joint velocities, and the Cartesian orientation and centre of mass) and action $u_t$ spaces are high-dimensional, i.e., $o_t \in \mathcal{R}^{41}$ and $u_t \in \mathcal{R}^8$.
> > > >
> > > > We use multi-task training with dropout to approximate the robot dynamics and compare different combinations of model-based controllers (Cross Entropy Method - CEM, Model Predictive Path Integral - MPPI, and Random Shooting - RS) and adaptation strategies (No adaptation, SMCD, and gradient descent) during an online control task. During evaluation, we randomly sample a leg to cripple on this quadrupedal robot. This task measures the ability of the proposed approach to work in high-dimensional spaces and adapt to an unexpected and drastic change to the underlying dynamics. We note that the approximate dynamic model is trained using samples in which all legs are functional.
> > > >
> > > > Average rewards obtained for a range of model-based RL control strategies, along with none (N), gradient (G), and SMCD (S) model adaptation are shown in the attached pdf. Both adaptation strategies perform similarly here, in and out of distribution. Importantly model adaptation does improve control performance, particularly when drastic changes in the underlying dynamics are introduced (right-plot). These results suggest that the proposed approach is also suitable for high-dimensional task spaces. However, it should be noted that there is significant confounding arising from the controller performance (no model-specific parameter tuning was performed for each controller) and the sampling strategy used to gather training data (all samples were obtained using randomly chosen actions), which may not adequately reflect the performance of the underlying adaptation strategies across all regions of the state space to the extent that the Panda control experiments test. It is likely that improved RL control and exploration strategies are needed to fully reap the benefits of SMCD.

---

### Official Review · Reviewer_iKY7 · 2022-07-25

**Originality:** Good
**Technical Quality:** Good
**Clarity Of Presentation:** Good
**Impact:** 2

**Recommendation:**

Weak Reject: I recommend rejecting the paper, but will not argue for my recommendation if the majority of other reviewers have a different opinion.

**Summary:**

This paper is considering the problem of adapting neural models to changes of task settings. The proposed method is to train a neural model with using dropout technique and then compute the most suitable dropout mask for the particular task at each step. The authors compare their approach with an oracle and another meta-learning algorithm called Reptile. The proposed method does not require any additional training of existing neural model and outperform more complex meta learning (Reptile) algorithm.

**Issues:**

* Describe the effect of the parameter N that stand for number of mask to use.
* Discuss how the method may / may not be used in real-world applications.

**Quality Of The Limitations Section:**

Limitations are addressed clearly

**Reviewer Expertise:**

2: The reviewer is willing to defend the evaluation, but it is quite likely that the reviewer did not understand central parts of the paper

**Robotics Focus:**

Highly relevant to robotics but no hardware experiments

**Strengths And Weaknesses:**

Strengths:
* The proposed method doesn’t require additional training of underlying neural model.
* Authors consider two different experiments.
* Outperforms the existing solution.

Weaknesses:
* It does not clear from the article how the number of mask N affect the method performance.
* Authors consider only simulated environments

**Summary Of Recommendation:**

This is an interesting paper with potential impact for robot online adaptation, however there is no evidence that this, method would be good enough in complex real world robotic problem.

---

> ### Author Response · Authors · 2022-08-18
> **Response to Reviewer iKY7**
>
> Thank you for your feedback and comments. We briefly respond to reviewer queries below and are currently running additional experiments to address concerns around the applicability of the proposed approach in complex robotic problems.
>
> **…not clear from the article how the number of mask N affect the method performance.**
> Thank you for pointing this out, as it is a strength of the proposed approach. We have added additional ablations showing that the proposed approach is not extremely sensitive to the number of masks required. This is to be expected, as bootstrap particle filters break the curse of dimensionality [1] (convergence is independent of state dimensionality, and the rate of convergence is in 1/N, where N here is the batch size or number of masks). We have added this important aspect to the paper. This information has been added to the supplementary material and page 4 - lines 151 to 160 in the main text.
>
> [1] D. Crisan and A. Doucet, "A survey of convergence results on particle filtering methods for practitioners," in IEEE Transactions on Signal Processing, vol. 50, no. 3, pp. 736-746, March 2002, doi: 10.1109/78.984773.
>
> **Authors consider only simulated environments**
>
> We note that although experiment 2 is in a simulated environment, data was gathered from real participants interacting with the simulator, and reflects a real-world behaviour modelling task.
>
> **This is an interesting paper with potential impact for robot online adaptation, however there is no evidence that this method would be good enough in complex real world robotic problem.**
>
> We are currently running additional experiments on a broader suite of control tasks (closer to complex real-world robotic problems) to better showcase the efficacy of the proposed approach.

---

> > ### Author Response · Authors · 2022-08-27
> > **Additional robotic experiments**
> >
> > We have added experiments on a 7-DOF Panda arm. In this experiment, a 7-DOF Panda manipulator is simulated with a tool that introduces an arbitrary unknown end-effector extension. Our goal is to drive the Panda from a fixed reference pose to within 5 cm of a randomly selected and kinematically feasible goal within 200 time-steps, using a PD control law and the Jacobian of the end-effector position joint with respect to the angles. We consider the case where measurement of the true 7-DOF Manipulator Control (Panda) end effector position is only available every nth timestep, which requires model-reference control when measurements are not available. This measures the ability of the adapted model to generalise to global context changes under more realistic control settings, where observations are captured at a lower rate than the required controller frequency.
> >
> > The success rates in the adaptive Panda control task for decreasing rates of measurement are shown in the attached image. FK is an analytical model that fails to account for the tool offset. When true measurements are available at every time step, this represents an oracle model and the only failures here are due to singularity effects in the control law. M+G is a multi-task model adapted using gradient descent, M+N is the same model with no adaptation, and M+S is the same model adapted using SMCD. M+G performance degrades significantly as the rate of measurement decreases and the model is required to be used for longer control periods. When selecting the M+G learning rate, we found that this performance degradation increases with the learning rate, which appears to indicate that gradient descent is only suitable to adapt representations to very local contexts, and the model is no longer useful in other regions of the state space. In contrast, SMCD appears to be much more effective at capturing global context, although performance still degrades when the model is used over longer periods with no feedback. Increasing the number of particles results in moderately increased performance.

---

### Official Review · Reviewer_PBaL · 2022-07-28

**Originality:** Good
**Technical Quality:** Fair
**Clarity Of Presentation:** Very Good
**Impact:** 2

**Recommendation:**

Weak Reject: I recommend rejecting the paper, but will not argue for my recommendation if the majority of other reviewers have a different opinion.

**Summary:**

This paper proposes a meta learning algorithm for adaptive robotic control in dynamic and unstructured environments.

The key idea is to first create a pre-trained neural net with dropout regularization (that was used during training) on multi-task data representing the entire task space. During the online phase, a posterior (or belief) over the dropout mask is being updated and propagated via a MCMC particle filtering scheme.

At each time step, the average of the propagated particles (representing the mask posterior) could be used to condition the pre-trained net (e.g. retaining most relevant neurons while dropping irrelevant neurons) to fit the current dynamic of a changing environment.

The experiment is conducted on two settings: (1) a toy setting where the task is to predict next state given current state & input control; and (2) a more realistic setting where the goal is to predict the (latent) landing skill level of human operators given their demonstrated landing sequences.

**Issues:**

I have pointed out several issues with this paper above. To summarize:

1. Lack of consideration regarding scalability, memory & compute complexity (in both technical narrative & experiment)
2. Experiment focuses on toy settings that do not match well with the motivation
3. Comparison with more recent baselines in meta learning is missing

While I do think the idea presented here is interesting, I am not quite convinced of its impact because of the above. I encourage the authors to expand the paper according to the above to complete & strengthen this work.

**Quality Of The Limitations Section:**

Limitations are addressed clearly

**Reviewer Expertise:**

3: The reviewer is fairly confident that the evaluation is correct

**Robotics Focus:**

Relevant but unlikely to deploy to hardware in near future

**Strengths And Weaknesses:**

Strength:

The idea of repurposing the dropout regularization as a mechanism for online adaptation of a pre-trained model is interesting.
The paper is well-motivated and has a comprehensive review of literature on model pre-training & adaptation techniques.
The mathematical exposition is neatly presented, communicating concisely the key idea via detailed pseudocode.

Weakness:

Although the idea is interesting, it has several restrictions:

1. First, given the formulation, it can only be applied to situations where all tasks in the space of interest share the same output space. This is much less flexible than other meta learning approach, such as MAML, which allows training & test tasks to focus on different output sub-spaces.

2. Second, generating pre-trained net on a master space of output & multi-task data that represents the entire task space is probably much more expensive in both compute and memory footprint.

For more sophisticated learning scenarios, the joint output space of the tasks could easily have tens of thousands of dimensions even though each task only focus on a subspace of output with less than a handful of dimensions. For such cases, it is arguable that a MAML model (whose output dimension is no more than that of any single task) is more cost-efficient & more suitable as an adaptation base.

3. Next, although the motivation of this paper highlight use cases where model adaptation is necessary to enable adaptive interaction to changes in environment or behavior of human collaborator (line 33), the actual reported experiments do not reflect such cases.

In fact, both experiments are not about learning and adapting control policies. Instead, one experiment is about predicting next state given acting policies while the other is about predicting intrinsic attributes of human operators given their demonstrated landing sequences. The authors should consider including dedicated experiments towards learning adaptive control policies, as these experiments are more relevant to what is being motivated.

4. Last, the comparison baseline is a bit limited to Reptile and does not include any of the more recent work in meta learning. The authors also mentioned that there was comparison with first-order MAML (line 205) but that does not seem to be included in the reported experiment. Moreover, there should also be discussion & comparison of processing complexity of the baselines. For example, for the proposed method to be effective, what would be the necessary sample size?

**Summary Of Recommendation:**

The paper proposes a seemingly interesting idea for meta learning via learning and propagating a posterior on dropout mask for a pre-trained net on a collective set of multi-task data. However, the authors had not considered practical aspects of such idea, e.g. scalability, memory & compute footprint etc. From the technical description, it is likely that the proposed method is significantly more costly than existing methods, at least in term of representation complexity.

In terms of experiments, there is a lack of comparison with more recent work in meta learning. The experiments also focus largely on toy dataset & in settings that are only remotely relevant to what was used to motivate this paper. Given this, I believe this paper is still a work in progress that unfortunately does not meet the bar for publication yet.

---

> ### Author Response · Authors · 2022-08-18
> **Response to Reviewer PBaL (Part 1)**
>
> Thank you for your feedback and comments. We respond to these below and address some misconceptions about the aims and claims of this work.
>
> **Less flexible than other meta learning approach, such as MAML**
>
> We agree that the core idea behind MAML is highly flexible and potentially more applicable to extremely diverse multi-task learning settings. However, we stress that this is not the objective of this paper - our goal is to introduce a new approach to updating models in an online sequential fashion and help solve a more targeted problem. As shown in our results, this is something that many more general meta-learning approaches struggle with. We will rephrase some of our statements around meta-learning to clarify these aspects and make it clear that we make no claims about introducing a new meta-learning approach.
>
> **Generating pre-trained net on a master space of output & multi-task data**
>
> We note a recently released work directly comparing multi-task training and fine tuning with meta learning in RL settings corroborates our findings [1]. This large scale study across a broad benchmark of meta-RL tasks concludes that multi-task training performs equally as well or better than meta-RL on similar levels of data. We suspect there are multiple trade-offs to be made here (memory vs compute and efficacy). It is also worth noting that meta-rl approaches need to train for longer to reach convergence. We have added this information to the supplementary material (lines 575-579).
>
> [1] Mandi, Zhao, Pieter Abbeel, and Stephen James. "On the Effectiveness of Fine-tuning Versus Meta-reinforcement Learning." arXiv preprint arXiv:2206.03271 (2022).
>
> **… the joint output space of the tasks could easily have tens of thousands of dimensions even though each task only focus on a subspace of output with less than a handful of dimensions**
>
> We are not sure we understand this point, could the reviewer clarify? Model output spaces are typically a fixed dimensional state or action vector.
>
> **It is arguable that a MAML model is more cost-efficient & more suitable as an adaptation base.**
>
> We note that the proposed adaptation approach could also be used with any model (even a MAML one), provided the model is trained with dropout.
>
> **Next, although the motivation of this paper highlight use cases where model adaptation is necessary to enable adaptive interaction to changes in environment or behavior of human collaborator (line 33), the actual reported experiments do not reflect such cases.**
>
> We will revise the motivation to be more clear about our intent. Experiments 4.2 show that we can adapt a human behaviour model and capture context-specific information using the inferred dropout masks. The aim of our work is not to introduce a new method for human-robot interaction, but rather to show that our approach adapts to and captures context-specific information. While it may be possible to construct an experiment where we use this within a robot policy, this doesn’t really add to the evaluation of the proposed approach, and we feel this would distract readers from the core contribution of this work - a novel and simple approach to online neural model adaptation that captures context-specific information.
>
> **The authors should consider including dedicated experiments towards learning adaptive control policies, as these experiments are more relevant to what is being motivated.**
>
> We are currently running additional experiments on a broader suite of control tasks to better showcase the efficacy of the proposed approach, along with adaptation algorithms more closely related to our problem setting than meta-learning techniques, which we acknowledge are not specifically designed to operate in these settings. We are primarily concerned with model-based control settings and note that the 2-link arm experiment is a model-based control task that shows how updating a model to improve predictions improves control convergence.

---

> > ### Author Response · Authors · 2022-08-18
> > **Response to Reviewer PBaL (Part 2)**
> >
> > **Lack of consideration regarding scalability, memory & compute complexity (in both technical narrative & experiment)**
> >
> > Thank you for pointing this out. We will add this discussion to the paper, as this is a particular strength of our work. We note that multi-task training with dropout is a standard training process and no more computationally expensive than any other general model training approach. Our results show that we achieve faster convergence with multi-task training than with reptile (fewer epochs) and [1] has shown improved performance with similar amounts of data. Note that our experiments used the same dataset for both reptile and multitask training.
> >
> > In terms of the memory requirements, our approach requires a single batch forward pass at inference time (N*model_params). In contrast, a gradient-based update rule requires (~2*model_params), but converges slowe. This greater memory footprint allows for a faster update scheme and more rapid adaptation.  We have added additional ablations showing that the proposed approach is not particularly sensitive to the number of masks required. This is to be expected, as bootstrap particle filters break the curse of dimensionality [2] (convergence is independent of state dimensionality, and the rate of convergence is in 1/N, where N here is the batch size or number of masks). There is existing theory on choosing N for filters of this form [3], and the formulation allows for variable batch sizes if desired, to allow even more efficient adaptation. As a result, we believe the proposed approach is highly scalable, particularly given the typical VRAM associated with modern GPUs, where batch sizes of 128 are entirely feasible for models of the size typically used in robot control.
> >
> > Information regarding the computational complexity of the proposed approach has been added to the supplementary material in the new section *Additional properties of SMCD*.
> >
> > [2] D. Crisan and A. Doucet, "A survey of convergence results on particle filtering methods for practitioners," in IEEE Transactions on Signal Processing, vol. 50, no. 3, pp. 736-746, March 2002, doi: 10.1109/78.984773.
> > [3] Kong, A, Liu, J.S. and Wong, W. H. (1994) Sequential imputations and Bayesian missing data problems. Journal of the American Statistical Association, 89, 1032–1044.
> >
> > **The authors also mentioned that there was comparison with first-order MAML (line 205) but that does not seem to be included in the reported experiment.**
> >
> > Our understanding is that Reptile is mathematically similar to a first order MAML approach. We will correct this to clarify.

---

### Official Review · Reviewer_J9eA · 2022-08-01

**Originality:** Good
**Technical Quality:** Fair
**Clarity Of Presentation:** Good
**Impact:** 3

**Recommendation:**

Weak Accept: I recommend accepting the paper, but will not argue for my recommendation if the majority of other reviewers have a different opinion.

**Summary:**

The ability of a model to adapt to online data can allow robots to achieve better and more robust performance when deployed in dynamic environments. Current methods, such as gradient-based meta-learning can be difficult to train and implement. In this work, the authors propose a novel adaptation framework for neural networks that is based on Monte Carlo Dropout. At train time, the network is trained using dropout (random masks applied at each forward pass). However, at test time, the authors do inference over dropout masks to find the best performing masks based on observed online data. This is done by particle filtering over masks using the proposed SMCD method. The authors show in two domains that their method leads to better prediction performance than baselines and further analyze the information content of the masks.

**Issues:**

Please see the weaknesses section.

**Quality Of The Limitations Section:**

Limitations are addressed clearly

**Reviewer Expertise:**

4: The reviewer is confident but not absolutely certain that the evaluation is correct

**Robotics Focus:**

Highly relevant to robotics but no hardware experiments

**Strengths And Weaknesses:**

This paper addresses an important and long-standing question in the robotics community: how can we adapt models efficiently based on online, observed data? One strength is the generality and capacity of the proposed approach that has potential to scale and model complex phenomena. Additionally, the approach is well-presented and straightforward to implement.

The main weaknesses of the paper in its current state relate to the “Experimental Results” section. These weaknesses can be divided into clarity and additional comparisons.

In terms of clarity, I believe addressing the following concerns would help the reader better understand the intuition/strengths behind the proposed approach:
- For the proposed SMCD algorithm, there are burn-in interactions to seed the adaptation process. Does the non-SMCD baselines observe the same total number of interactions before prediction error is reported?
- It would be beneficial to have more details about the Reptile baseline algorithms. Specifically, how does the R+S baseline work, where Reptile is combined with SMCD work?
- In Figures 3 and 5, why does M-N outperform R-N? Without adaptation, what are the differences between these models that would explain this performance gap?
- In Figure 3, why does R-N outperform R-G? More details on the Reptile baseline might help answer this.

The authors provide many insightful ablation studies to their proposed algorithm (for example, the no-adaptation baseline, or the oracle model). As the authors develop a novel framework for adaptive model learning, these ablations are important to understand where performance gains come from and I believe the following ablations would provide further insight:
- As I understand, in the two-link arm domain, the networks do not have access to the link length and are adapting to this unknown task context. A baseline model that has no adaptation, but has access to all relevant state as input would be a useful benchmark as I would expect it to be an upper bound on performance.
- There are methods in the literature that explicitly capture variability using a latent variable, as opposed to the proposed method which captures task variability using all the weights in the network combined with dropout masks. A comparison with this class of models would also add insight (does the proposed approach lead to better performance by using the entire model to capture task variability instead of a low-dimensional latent-variable?). One approach in this category is that of Saemundsson et al., 2018 (“Meta reinforcement learning with latent variable gaussian processes”).

The following are minor comments:
- Is there any intuition why a mean-mask estimator in Equation 6 works well? Would you expect this distribution to be unimodal?
- I would be interested in reading any additional insights into how the network specializes for different contexts (as shown by the dropout mask analysis) even though random masks are used at training time
- What are the units in Figure 3?

**Summary Of Recommendation:**

In the current state, I recommend a “Weak Reject” for this paper. The authors tackle an interesting problem and propose a novel framework for learning adaptive models. However, additional clarity is required to fully interpret the strengths and weaknesses of the proposed method. I look forward to the authors’ responses as I believe many of the identified concerns are addressable.

Post-Discussion Update:
I would like to thank the reviewers for taking the time to respond to all the points in my review. I believe that the additional clarity greatly improves the paper and have updated my score to a weak accept. In particular, the explanation provided for why R-N outperforms M-N is very intuitive.The additional experiments also provide a stronger set of baseline methods.

---

### Meta-Review · Area_Chair_9xMx · 2022-08-12

**Recommendation:** Accept (Poster)
**Confidence:** 3

**Metareview:**

Tackling the important problem of online adaptation with the proposed dropout mask adaptation during runtime is a simple but clever idea, as it requires no additional training, is easy to use, and could potentially be transferred to many domains and models. The approach is well motivated and presented.

However, there are some open questions and concerns raised by the reviewers. Please see the following proposed action items based on these (more details in the corresponding reviews), and the consensus is that most concerns are addressable and hence the authors are encouraged to think about investigating them.

Proposal of action items:
- Clarification of the burn-in interactions with respect to the baseline models and reported prediction errors
- Clarification of some of the reported results and models (R+S, M-N vs. R-N, R-N vs. R-G, see review)
- Consider additional ablations and comparisons for more insights
- Consider more experiments with respect to learning adaptive control strategies, higher-dimensional systems, different underlying models, etc.. , to create a strong message for the proposed approach (see review for proposed environments)
- Clarify practical aspects of the approach, e.g. scalability, memory & compute footprint, etc.
- Clarifying the effect of the number of masks N affect performance

-----

I’d to thank all reviewers and authors for constructive feedback, clarifications, and additional material. There is consensus that the presented method tackles an important problem and that the method is well motivated and presented. While there is no clear consensus in the recommendations, the updated and significantly enriched paper (in addition to the rebuttal discussions), tackled most of the raised concerns. My recommendation is to accept the paper due to its strengths that outweigh the flaws. The proposed idea is clever and simple, easy to use on a variety of settings and models, and has the potential to spark many new ideas in this direction.


**Best Paper Nomination:**

No

---

> ### Author Response · Authors · 2022-08-27
> **Response to Area Chair 9xMx**
>
> We wish to thank all reviewers and the AC for feedback that we believe has resulted in a significantly improved paper. We briefly respond to the action items above.
>
> - All models had equivalent burn-in interactions
> - Results have been clarified and explanations provided
> - We have included additional ablations in the appendix (number of particles, likelihood uncertainty, transition probability)
> - We have added experiments on a 7-DOF Panda arm and a Crippled Ant environment (49 dim) which provide important insights into scalability to higher dimensions, extrapolation to unseen environments, and practical applicability in real-time control. We have also added additional baselines (analytical models, latent variable v-rnn approaches). We are currently working on further improving the v-rnn baseline.
>
> We have clarified practical aspects:
> - Multitask training with dropout is cheaper than double loop learning
> - Particle filters break the curse of dimensionality (convergence independence of state dim, in 1/N, thus highly scalable)
> - The proposed approach is embarrassingly parallel, requires a single forward pass and resampling step (rapid inference)
> - SMCD requires additional memory footprint, but not excessive, almost all models in this work could be fit in VRAM on a Jetson Nano Embedded GPU
>
> We have added additional experiments to showcase the effect of N (ablations/ Panda arm) - performance increases with the number of masks. Still, since the particle filter breaks the curse of dimensionality (convergence independence of state dim, in 1/N) this is not excessive. We have also added citations to the theory on selecting N, which can be adjusted online by computing the effective sample size.
> These additional insights have also highlighted the strengths of the proposed approach. It is simple to implement and broadly applicable to any model trained with dropout. Adaptation of the masks using SMCD can be done rapidly and online, and experiments show that inferred context masks capture more global information than approaches that adjust model representations using gradient descent. In an online setting, the latter shows catastrophic forgetting and overly local adaptation, while SMCD preserves the representation and produces a model capable of longer horizon predictions. The inferred masks capture valuable context, which is of interest in human behaviour modelling and opens up downstream applications in collaborative control. As mentioned by all reviewers, these results are novel and interesting and address an important challenge of significant interest to the robot learning community.